# Comparison of clinical outcomes of angiotensin receptor blockers with angiotensin-converting enzyme inhibitors in patients with acute myocardial infarction

**Chih-Wei Chen**[1,2,3☯], **Chun-Wei Chang**[1,2☯], **Yi-Cheng Lin**[4,5], **Wan-Ting Chen**[6], **Li-Nien Chien**[6,7‡]*, **Chun-Yao Huang**[1,2,3‡]*

**1** Division of Cardiology, Department of Internal Medicine and Cardiovascular Research Center, Taipei Medical University Hospital, Taipei, Taiwan, **2** Taipei Heart institute, Taipei Medical University, Taipei, Taiwan, **3** Division of Cardiology, Department of Internal Medicine, School of Medicine, College of Medicine, Taipei Medical University, Taipei, Taiwan, **4** Department of Pharmacy, Taipei Medical University Hospital, Taipei, Taiwan, **5** School of Pharmacy, College of Pharmacy, Taipei Medical University, Taipei, Taiwan, **6** Institute of Health and Welfare Policy, College of Medicine, National Yang Ming Chiao Tung University, Taipei, Taiwan, **7** Graduate Institute of Data Science, College of Management, Taipei Medical University, Taipei, Taiwan

☯ These authors contributed equally to this work.
‡ LNC and CYH also contributed equally to this work.
* linien.chien@nycu.edu.tw (LNC); cyhuang@h.tmu.edu.tw (CYH)

**Data Availability Statement:** All relevant data are within the paper and its Supporting information files.

## Abstract

### Background

Angiotensin receptor blockers (ARBs) are considered an alternative to angiotensin-converting enzyme inhibitors (ACEIs) in patients with acute myocardial infarction (AMI), but in the era of extensive use of preventive therapies and percutaneous coronary intervention, this has not been adequately evaluated in Asians.

### Methods

This retrospective cohort study used data from the Taiwan National Health Insurance Research Database. In total, 52,620 patients initially hospitalized due to AMI between 2002 and 2015 were assessed.

### Results

After propensity score matching, 14,993 patients each were assigned to ACEI and ARB groups. Patients who received ARBs had significantly lower all-cause mortality (adjusted hazard ratio [aHR]: 0.82; 95% confidence interval [CI]: 0.75–0.90) and hospitalization for heart failure (aHR: 0.92; 95% CI: 0.85–0.99) compared with those who received ACEIs at 18 month follow-up. No significant difference was observed between the two groups in terms of major adverse cardiovascular events (aHR: 098; 95% CI: 0.90–1.07), cardiovascular death (aHR: 0.82; 95% CI: 0.68–1.00), ischemia stroke (aHR: 0.93; 95% CI: 0.77–1.11),

**Funding:** This study was supported by the National Science and Technology Council of Taiwan in the form of a sponsorship to C-YH [NSTC-111-2218-E 008-009].

**Competing interests:** The authors have declared that no competing interests exist.

and nonfatal myocardial infarction (aHR: 1.04; 95% CI: 0.93–1.17). ARBs showed benefits in many subgroups in terms of all-cause mortality and cardiovascular death.

## Conclusions

Real-world data demonstrate that ARBs might be associated with lower all-cause mortality and hospitalization for heart failure compared with ACEIs among patients with AMI.

## Introduction

Angiotensin-converting enzyme inhibitors (ACEIs) and angiotensin receptor blockers (ARBs) have been recommended as preventive therapies for patients with acute myocardial infarction (AMI) in the European Society of Cardiology and American College of Cardiology/American Heart Association (ACC/AHA) guidelines [1–3]. ACEIs were recommended as standard treatment in patients with myocardial infarction (MI), particularly in patients with left ventricular systolic dysfunction, diabetes, or chronic kidney disease. By contrast, ARBs are considered alternatives to ACEIs when patients are intolerant to ACEIs [1–3]. The prescription rate of ARBs has prominently increased worldwide over the past several years [4, 5]. This phenomenon is significant in Asian countries because persistent cough is a common side effect associated with ACEI therapy among Eastern Asians [6, 7]. Many trials have demonstrated the beneficial roles of ACEI in patients with AMI [8, 9], and ARBs have been found to be noninferior to ACEIs in patients with AMI for major adverse cardiovascular event (MACE) outcomes [10, 11]. However, ARBs have not been adequately compared with ACEIs in Asians with AMI, particularly in the modern era with extensive use of statins and timely percutaneous coronary intervention (PCI). Additionally, the use of ACEI or ARB could result in acute kidney injury (AKI) and hyperkalemia; thus, adverse events are common in patients with chronic kidney disease (CKD) [12, 13]. Furthermore, evidence of renal outcomes after ACEI/ARB treatment of patients with AMI, followed by multiple comorbidities, is rare.

Therefore, we conducted this study to investigate the effectiveness of ACEI and ARB in terms of mortality, MACE, hospitalization for heart failure (HHF), and renal outcomes in Asians with AMI by using the data from real-world settings.

## Material and methods

### Ethics statement

This study was approved by the Joint Institutional Review Board of Taipei Medical University (TMU-JIRB No. 201911004). The need for informed consent was waived owing to the use of anonymized data.

### Study design and data source

This study adopted a retrospective cohort design and used data files from National Health Insurance Research Database (NHIRD) in Taiwan. NHIRD is a population-based claims database of the National Health Insurance (NHI) program, a single-payer health insurance program initiated in 1995. NHI provides a comprehensive coverage in Taiwan, including outpatient visits, the inpatient system, drug prescriptions, treatment with traditional Chinese medicine, dental services, operations, and examinations through radiographic and magnetic resonance images. Under the legislation, all legal residents of Taiwan are eligible for NHI

benefit and are required to enroll for the program; therefore, the coverage of NHI reached 99.9% of the Taiwanese population by the end of 2017. Vital status and cause of death were ascertained from the National Death Registry of Taiwan. The two datasets can be linked by a unique encrypted identifier under the regulation of Health and Welfare Data Science Center, Ministry of Health and Welfare, Taiwan.

## Study cohort

Patients hospitalized with a diagnosis of AMI between 2002 and 2015 were first selected, and the date of first AMI admission was treated as the index date of AMI. Those patients' data was first accessed on December 1st, 2019. To confirm the diagnosis, only patients who received heparin or antiplatelet agents during their AMI admission were included in the study. We excluded patients (1) aged <20 years, (2) with missing sex information, (3) who were not a citizen of Taiwan, (4) who had undergone coronary artery bypass or had a history of AMI (a 2-year washout period), (5) had a death record or AMI or stroke hospitalization within 3 months after the index date of AMI, and (6) had a dialysis claims before or within 3 months after the index date of AMI. Moreover, we excluded patients who were hospitalized for >1 month because it indicated that their disease condition was complicated.

Therefore, we additionally excluded patients who did not receive ACEI or ARB or received both ACEI and ARB or switched one medication to another or had a medication possession ratio (MPR) of <0.4 within 3 months after the index date of AMI. The MPR was determined based on the proportion of days on medication within 3 months. Finally, the patients were classified into the ACEI or ARB group within 3 months after the index date of AMI. The detailed flow chart of patient selection is presented in Fig 1.

## Propensity score matching

To reduce the potential selection bias, we used one-to-one propensity score matching (PSM) to select a pair of patients who had similar baseline characteristics but different treatments (either ACEI or ARB), so that the two groups could have similar distributions (to a comparable degree) of observed baseline covariates. We applied the nearest neighbor matching within 0.1 caliper distance of propensity score to select patients and considered the covariates of age, sex, diagnostic year, comorbidities, and medication use to estimate the propensity score (Table 1). A logistic regression analysis was used to calculate propensity scores. The similarity between the two groups was evaluated through standardized mean difference (SMD), and a SMD of <10% (or 0.1) indicated a negligible correlation between the treatment groups and variables.

The underlying comorbidities of participants included vascular comorbidities such as coronary artery disease (CAD), hypertension, diabetes, hyperlipidemia, peripheral arterial occlusive disease, heart failure, stroke, atrial fibrillation, and ventricle diseases; kidney diseases such as hyperuricemia and chronic kidney disease; and a history of bleeding. Medication use both before and after AMI were considered, including antiplatelet agents, clopidogrel, ACEI/ARB, statin, acetylsalicylic acid/dipyridamole, diuretics, calcium channel blockers, and beta-blockers. The International Classification of Diseases (ICD), Ninth Edition, Clinical Modification and ICD, Tenth Edition for disease diagnostic codes and the Anatomical Therapeutic Chemical code for medications are listed in S1 Table. Finally, PCI, IAPD, and extracorporeal membrane oxygenation use before the index date of AMI were considered to adjust the baseline difference between the two groups.

**Fig 1. Patient selection process.** Abbreviations: AMI = acute myocardial infarction; ACEI = angiotensin-converting enzyme inhibitor; ARB = angiotensin receptor blocker; CABG = coronary artery bypass; MPR = medication possession ratio.

## Main outcome measurements

Outcomes of treatment effectiveness were all-cause mortality, MACE, and HHF. MACE was defined as cardiovascular (CV) death, nonfatal myocardial infraction, and nonfatal ischemic stroke during the follow-up periods. Death records were derived from the National Death Registry, which provided the death date and death cause. Nonfatal myocardial infraction, stroke, and HHF were derived based on discharge records from NHIRD. Safety outcomes were AKI and renal failure. AKI was defined as a discharge diagnosis of AKI in conjunction with a procedure code of dialysis. Dialysis was defined as receiving dialysis continuously for 3 months (S1 Table).

## Statistical analysis

Descriptive statistics was used to present the characteristics of the ACEI and ARB groups. This study was mainly designed as an intention-to-treat analysis that ignores noncompliance and drug switching or drug withdrawal after enrollment. We believe that this approach can provide a good estimate of treatment efficiency when patients with a high adherence level are included. Additionally, we provided an as-treated analysis as a sensitivity analysis, in which the data of patients were censored if they switched or discontinued their initial medications.

Survival analysis along with Kaplan–Meier method and a stratified Cox proportional hazard regression was used to compare the outcome risk between ACEI and ARB. The follow-up periods were set at 12 and 18 months. Patients were followed from 3 months after the index date of

**Table 1. Baseline characteristics of new AMI survivors who received either ACEI or ARB within 3 months after discharge.**

| | Before PSM | | | After PSM | | |
|---|---|---|---|---|---|---|
| | ACEI | ARB | | ACEI | ARB | |
| | (%) | (%) | SMD | (%) | (%) | SMD |
| Sample size, n | 27,329 | 25,291 | | 14,993 | 14,993 | |
| Male | 82.1 | 71.3 | 0.256 | 81.5 | 81.5 | <0.001 |
| Age, years | | | | | | |
| Mean (SD) | 61.83 (13.85) | 65.36 (13.66) | 0.257 | 62.92 (13.13) | 62.92 (13.14) | 0.001 |
| 20–44 | 10.6 | 7.0 | 0.128 | 8.2 | 8.1 | 0.002 |
| 45–64 | 47.6 | 40.0 | 0.152 | 46.8 | 46.8 | <0.001 |
| 65–74 | 20.5 | 23.6 | 0.074 | 22.7 | 22.5 | 0.004 |
| ≥75 | 21.4 | 29.4 | 0.186 | 22.3 | 22.5 | 0.005 |
| DX year | | | | | | |
| 2002–2005 | 26.7 | 13.3 | 0.339 | 16.4 | 16.4 | <0.001 |
| 2006–2010 | 38.2 | 32.7 | 0.116 | 35.4 | 35.4 | <0.001 |
| 2011–2015 | 35.2 | 54.1 | 0.387 | 48.2 | 48.2 | <0.001 |
| STEMI | 74.9 | 60.3 | 0.315 | 71.0 | 66.6 | 0.095 |
| History of cardiac procedure, yes | | | | | | |
| Prior PCI | 1.3 | 2.7 | 0.107 | 1.6 | 1.7 | 0.006 |
| IABP | 3.6 | 3.8 | 0.014 | 4.0 | 4.0 | 0.003 |
| ECMO | 0.2 | 0.2 | 0.015 | 0.2 | 0.2 | 0.003 |
| Comorbidities, yes | | | | | | |
| CAD | 67.2 | 73.9 | 0.149 | 74.8 | 72.6 | 0.050 |
| Hypertension | 62.8 | 79.0 | 0.362 | 69.5 | 70.3 | 0.016 |
| Diabetes | 33.3 | 41.1 | 0.163 | 35.0 | 34.5 | 0.010 |
| Hyperlipidemia | 51.7 | 52.8 | 0.023 | 53.3 | 52.5 | 0.016 |
| PAOD | 2.4 | 3.3 | 0.054 | 2.6 | 2.7 | 0.006 |
| Heart failure | 44.7 | 51.7 | 0.139 | 46.4 | 47.2 | 0.016 |
| Stroke | 6.8 | 8.4 | 0.062 | 6.8 | 6.6 | 0.008 |
| Atrial fibrillation or flutter | 5.8 | 7.9 | 0.085 | 6.1 | 6.1 | 0.001 |
| Ventricle tachycardia | 2.3 | 2.1 | 0.012 | 2.3 | 2.3 | 0.004 |
| Ventricular fibrillation or flutter | 1.5 | 1.3 | 0.016 | 1.4 | 1.5 | 0.007 |
| Gout or hyperuricemia | 11.5 | 12.0 | 0.013 | 11.0 | 12.1 | 0.033 |
| Intracerebral hemorrhage | 0.6 | 0.7 | 0.012 | 0.6 | 0.6 | 0.003 |
| Gastrointestinal bleeding | 5.1 | 6.0 | 0.039 | 5.0 | 4.9 | 0.004 |
| Other noncritical site bleeding | 1.7 | 2.1 | 0.029 | 1.9 | 1.9 | 0.001 |
| CKD | 3.8 | 7.7 | 0.172 | 4.4 | 4.3 | 0.004 |
| CLD | 11.0 | 15.0 | 0.119 | 11.6 | 12.6 | 0.031 |
| Cancer | 3.0 | 4.1 | 0.059 | 3.6 | 3.4 | 0.010 |
| Medication use, yes | | | | | | |
| Aspirin | 17.1 | 25.0 | 0.193 | 17.9 | 17.9 | 0.001 |
| Clopidogrel | 2.0 | 4.2 | 0.128 | 2.2 | 2.2 | 0.005 |
| Ticlopidine | 0.9 | 1.2 | 0.028 | 0.9 | 0.8 | 0.013 |
| CCB | 23.2 | 35.7 | 0.277 | 26.5 | 26.8 | 0.008 |
| Insulin | 2.8 | 5.7 | 0.146 | 3.1 | 3.2 | 0.005 |
| PPIs | 2.4 | 3.3 | 0.053 | 2.6 | 2.5 | 0.005 |
| Warfarin | 0.8 | 1.2 | 0.038 | 0.8 | 0.8 | <0.001 |
| NOACs | 0.0 | 0.1 | 0.030 | 0.0 | 0.0 | 0.010 |
| NSAIDs | 35.1 | 38.2 | 0.065 | 35.4 | 35.8 | 0.008 |

(*Continued*)

**Table 1.** (Continued)

| | Before PSM | | | After PSM | | |
|---|---|---|---|---|---|---|
| | ACEI | ARB | | ACEI | ARB | |
| | (%) | (%) | SMD | (%) | (%) | SMD |
| Urate-lowering agent | 8.7 | 11.1 | 0.083 | 8.7 | 9.4 | 0.024 |
| ACEI | 14.6 | 8.5 | 0.192 | 10.5 | 10.0 | 0.017 |
| ARB | 10.9 | 38.2 | 0.667 | 15.9 | 16.5 | 0.017 |
| Beta-blocker | 18.7 | 28.4 | 0.229 | 20.4 | 20.7 | 0.007 |
| Statins | 11.3 | 19.7 | 0.234 | 13.2 | 12.8 | 0.015 |
| **Medication use within 3 months after the index AMI discharge** | | | | | | |
| Aspirin | 91.8 | 88.9 | 0.098 | 91.8 | 91.4 | 0.014 |
| Clopidogrel | 77.8 | 77.5 | 0.008 | 79.2 | 78.9 | 0.007 |
| Ticagrelor | 6.7 | 10.5 | 0.133 | 10.5 | 8.8 | 0.058 |
| Ticlopidine | 2.6 | 1.6 | 0.069 | 1.4 | 1.7 | 0.019 |
| Beta-blocker | 70.1 | 69.7 | 0.008 | 70.9 | 70.1 | 0.017 |
| Statins | 62.4 | 66.3 | 0.082 | 68.1 | 65.7 | 0.052 |
| MPR within 90 days | | | | | | |
| ≥0.80 | 60.0 | 57.3 | 0.054 | 56.3 | 59.0 | 0.055 |
| 0.4–0.8 | 40.0 | 42.7 | 0.054 | 43.7 | 41.0 | 0.055 |

Abbreviation: AMI = acute myocardial infarction; ACEI = angiotensin-converting enzyme inhibitors; ARB = angiotensin receptor blocker; CAD = chronic artery disease; CCB = calcium channel blocker; CKD = chronic kidney disease; CLD = chronic lung disease; ECMO = extracorporeal membrane oxygenation; IABP = intra-aortic balloon pump; MPR = medication possession ratio; NOACs = novel oral anticoagulants; NSAIDs = nonsteroidal anti-inflammatory drugs; PAOD = peripheral arterial occlusive disease; PCI = percutaneous coronary intervention; PPIs = proton pump inhibitors; PSM = propensity score matching; SMD = standardized mean difference; SD = standard deviation. STEMI = ST segment elevation myocardial infarction.

AMI to whichever outcome occurred first: (1) the outcome date; (2) the death date; (3) end of the observation period; or (4) December 31, 2017. The cumulative event rates of interest were estimated using the Kaplan–Meier method. Furthermore, the incidence of events along with the 95% confidence interval (CI) was computed. A stratified Cox proportional hazard regression was used to compare the event risk between ACEI and ARB. The assumption of proportional hazards was assessed. All analyses were performed using SAS/STAT 9.4 software (SAS Institute Inc., Cary, NC, USA), STATA 14 software (Stata Corp LP, College Station, TX, USA), and R (version 3.2.5 for Windows). A value of $P < 0.05$ was considered significant.

## Results

### Baseline patient characteristics

We identified 52,620 patients who survived AMI after receiving either ACEIs or ARBs between January 2002 and December 2015. After PSM, 14,993 patients each were assigned to ACEI and ARB arms. As shown in Table 1, the rate of ARB prescription increased from 13.3% in 2002–2005 to 54.1% in 2011–2015 among patients with AMI. By contrast, the prescription rate of ACEIs increased slightly from 26.7% in 2002–2005 to 35.2% in 2011–2015. Before PSM, patients receiving ARBs were older and more likely to have comorbidities such as CAD, hypertension, diabetes mellitus, heart failure, and chronic kidney disease than those receiving ACEI. However, these intergroup differences in baseline characteristics were well-balanced after PSM, with the SMDs being <0.1 for all characteristics. The detailed baseline characteristics, underlying comorbidities, medication histories, and in-hospital treatment characteristics are presented in Table 1.

## Adherence of treatment group

Adherence to therapy was defined as a MPR of >40% within 3 months after the index date. As a result, 60.0% and 57.3% of ACEI and ARBs users, respectively, had a MPR of >0.8 within 90 days after the index date of AMI. During follow up, 8015 (53.4%) ACEI users switched to ARBs, and the median time of switching was 14.4 (6.3–36.7) months. By contrast, 3456 (23.1%) ARBs users switched to ACEIs during this period, and the median time of switching was 25.2 (10.1–51.1) months (S2 Table).

## Clinical outcomes

During the follow-up period, 807 and 641 ACEI and ARB users, respectively, died. For all-cause mortality at 18-month follow-up, the ARB group had significantly lower all-cause mortality than the ACEI group (adjusted hazard ratio [aHR]: 0.82; 95% CI: 0.75–0.90; Table 2). Kaplan–Meier survival estimates for 18-month all-cause mortality in matched cohorts are presented in Fig 2A. The trend of all-cause mortality after AMI decreased from 2002 to 2015, which is demonstrated according to the year of AMI diagnosis in Fig 2B. No significant difference was observed between the ARB and ACEI groups in terms of MACE (aHR: 0.98; 95%

**Table 2. Incidence (per 1000 PM) and adjusted hazard risk of the major outcomes for treatment efficiency based on intention-to-treat analysis.**

| Outcomes | Follow-up period | Treatment | Number of events | PM | Incidence (95% CI) | Adjusted* HR (95% CI) | P |
|---|---|---|---|---|---|---|---|
| **All-cause death** | 12M | ACEI | 807 | 174,689 | 4.62 (4.31–4.95) | 1.00 (Ref.) | |
| | | ARB | 641 | 175,783 | 3.65 (3.37–3.94) | 0.79 (0.71–0.87) | <0.001 |
| | 18M | ACEI | 1118 | 258,867 | 4.32 (4.07–4.58) | 1.00 (Ref.) | |
| | | ARB | 915 | 261,068 | 3.5 (3.28–3.74) | 0.82 (0.75–0.90) | <0.001 |
| **MACE** | 12M | ACEI | 798 | 170,586 | 4.68 (4.36–5.01) | 1.00 (Ref.) | |
| | | ARB | 764 | 171,827 | 4.45 (4.14–4.77) | 0.95 (0.86–1.05) | 0.334 |
| | 18M | ACEI | 1020 | 251,221 | 4.06 (3.81–4.32) | 1.00 (Ref.) | |
| | | ARB | 993 | 253,430 | 3.92 (3.68–4.17) | 0.98 (0.90–1.07) | 0.677 |
| **CV death** | 12M | ACEI | 188 | 174,689 | 1.08 (0.93–1.24) | 1.00 (Ref.) | |
| | | ARB | 148 | 175,783 | 0.84 (0.71–0.99) | 0.76 (0.61–0.95) | 0.016 |
| | 18M | ACEI | 240 | 258,867 | 0.93 (0.81–1.05) | 1.00 (Ref.) | |
| | | ARB | 203 | 261,068 | 0.78 (0.67–0.89) | 0.82 (0.68–1.00) | 0.05 |
| **Ischemic stroke** | 12M | ACE-I | 182 | 173,646 | 1.05 (0.90–1.21) | 1.00 (Ref.) | |
| | | ARB | 178 | 174,705 | 1.02 (0.87–1.18) | 0.92 (0.74–1.14) | 0.448 |
| | 18M | ACEI | 259 | 256,809 | 1.01 (0.89–1.14) | 1.00 (Ref.) | |
| | | ARB | 249 | 258,924 | 0.96 (0.85–1.09) | 0.93 (0.77–1.11) | 0.408 |
| **AMI** | 12M | ACEI | 498 | 171,544 | 2.90 (2.65–3.17) | 1.00 (Ref.) | |
| | | ARB | 501 | 172,811 | 2.90 (2.65–3.16) | 1.02 (0.90–1.16) | 0.722 |
| | 18M | ACEI | 618 | 253,106 | 2.44 (2.25–2.64) | 1.00 (Ref.) | |
| | | ARB | 625 | 255,388 | 2.45 (2.26–2.65) | 1.04 (0.93–1.17) | 0.501 |
| **HHF** | 12M | ACEI | 1273 | 166,756 | 7.63 (7.22–8.07) | 1.00 (Ref.) | |
| | | ARB | 1174 | 168,448 | 6.97 (6.58–7.38) | 0.90 (0.82–0.97) | 0.009 |
| | 18M | ACEI | 1530 | 244,556 | 6.26 (5.95–6.58) | 1.00 (Ref.) | |
| | | ARB | 1436 | 247,420 | 5.8 (5.51–6.11) | 0.92 (0.85–0.99) | 0.026 |

*Adjusted HR was estimated through stratification Cox regression adjusted for the covariates listed in Table 1.

Abbreviations: AMI = acute myocardial infarction; ACEI = angiotensin-converting enzyme inhibitor; ARB = angiotensin receptor blocker; CI = confidence interval; HHF = hospitalization for heart failure; HR = hazard ratio; M = month; MACE = major adverse cardiovascular events; PM = person month; Ref. = reference; CV = cardiovascular.

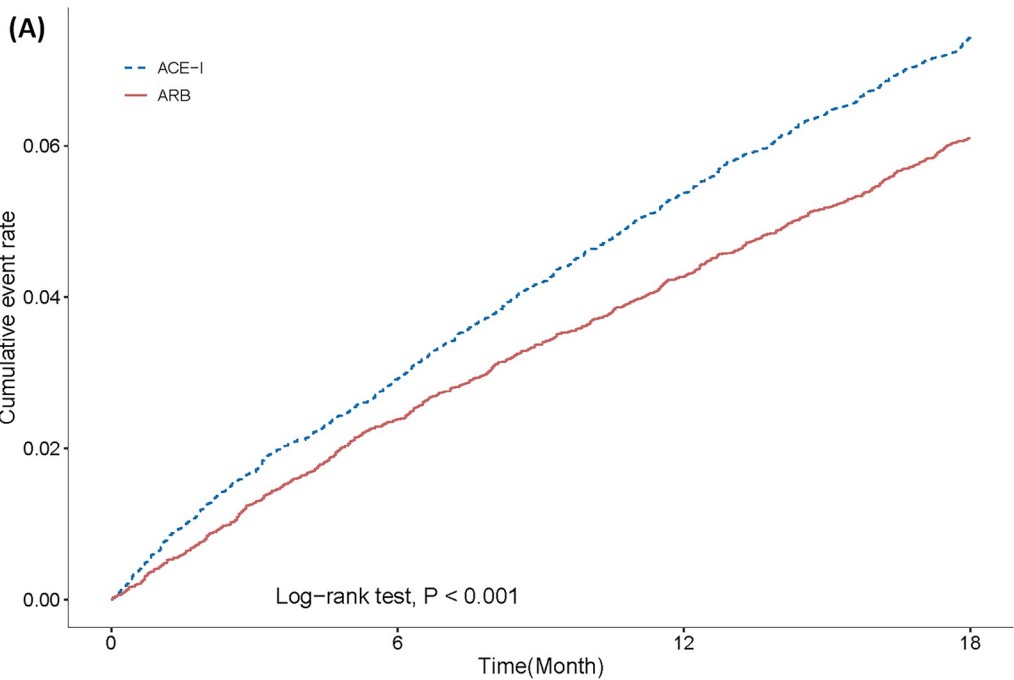

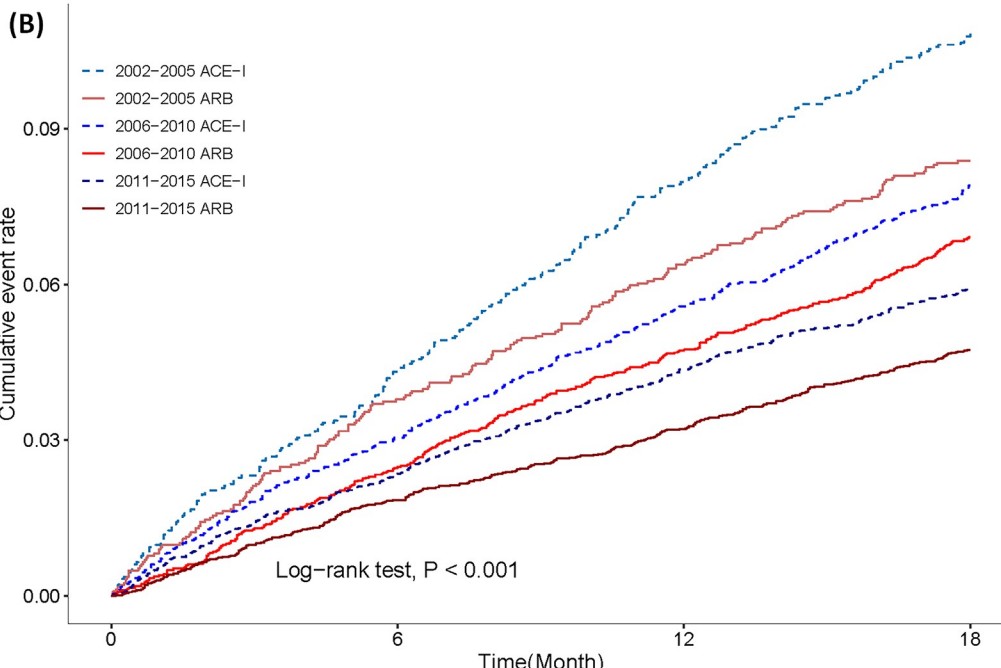

**Fig 2. Kaplan–Meier failure curve for all-cause death based on (A) treatment and (B) treatment and diagnostic year.**
Abbreviations: ACEI = angiotensin-converting enzyme inhibitors; ARB = angiotensin receptor blocker.

**Table 3. Incidence (per 1000 PM) and adjusted hazard risk of renal outcomes for treatment safety based on intention-to-treat analysis.**

| Outcomes | Follows-up period | Treatment | Number of Events | PM | Incidence (95% CI) | Adjusted* HR (95% CI) | P |
|---|---|---|---|---|---|---|---|
| **AKI** | 12M | ACEI | 255 | 173,576 | 1.47 (1.29–1.66) | 1.00 (Ref.) | |
| | | ARB | 238 | 174,724 | 1.36 (1.19–1.55) | 0.93 (0.77–1.11) | 0.408 |
| | 18M | ACEI | **339** | **256,788** | **1.32 (1.18–1.47)** | 1.00 (Ref.) | |
| | | ARB | 317 | 259,035 | 1.22 (1.09–1.37) | 0.93 (0.79–1.09) | 0.355 |
| **Dialysis** | 12M | ACEI | 68 | 174,361 | 0.39 (0.30–0.49) | 1.00 (Ref.) | |
| | | ARB | 89 | 175,407 | 0.51 (0.41–0.62) | 1.19 (0.86–1.65) | 0.284 |
| | 18M | ACEI | **98** | **258,207** | **0.38 (0.31–0.46)** | 1.00 (Ref.) | |
| | | ARB | 119 | 260,248 | 0.46 (0.38–0.55) | 1.14 (0.86–1.50) | 0.36 |

*Adjusted HR was estimated through stratification Cox regression adjusted for the covariates listed in Table 1.

Abbreviations: AKI = acute kidney disease; ACEI = angiotensin-converting enzyme inhibitor; ARB = angiotensin receptor blocker; CI = confidence interval; HR = hazard ratio; M = month; PM = person month; Ref. = reference.

CI: 0.90–1.07), CV death (aHR: 0.82; 95% CI: 0.68–1.00), ischemia stroke (aHR: 0.93; 95% CI: 0.77–1.11), and non-fatal MI (aHR: 1.04; 95% CI: 0.93–1.17) at 18-month follow-up. However, HHF in the ARB group were significantly lower than in the ACEI groups (aHR: 0.92; 95% CI: 0.85–0.99).

For renal outcomes, no significant difference was observed between the two groups in terms of AKI (aHR: 0.93; 95% CI: 0.79–1.09) and dialysis (aHR: 1.14; 95% CI: 0.86–1.50) at 18-month follow-up (Table 3). The subgroups did not show heterogeneity in the effect of treatment on the risk of 18-month mortality and CV death (Fig 3).

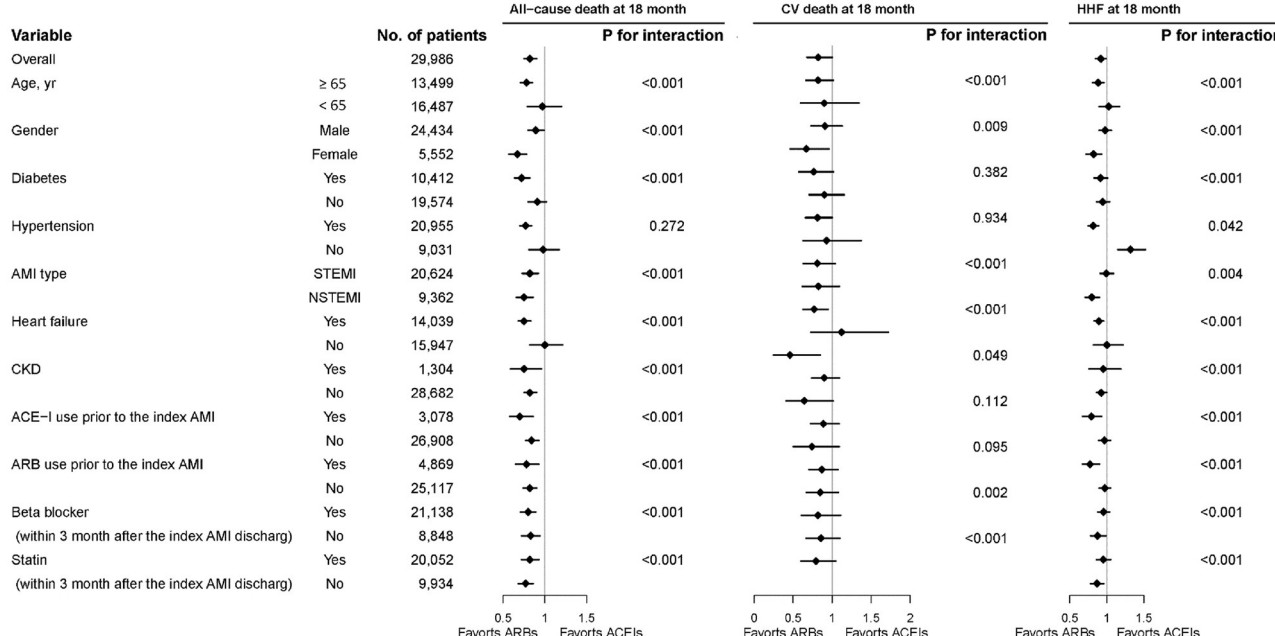

**Fig 3. Subgroup analysis of the hazard risks of all-cause cause, CV death, and HHF within an 18-month follow-up period based on intention-to-treat analysis.** Abbreviations: AMI = acute myocardial infarction; ACEI = angiotensin-converting enzyme inhibitor; ARB = angiotensin receptor blocker; CV = cardiovascular; HHF = hospitalization for heart failure; HR = hazard ratio; M = month; PM = person month; Ref. = reference; CKD = chronic kidney disease.

We also performed a sensitivity analysis in which the data of patients were censored if they switched the initial medication. The results were consistent with the main findings, as shown in S3 and S4 Tables.

## Discussion

In this nationwide cohort study of survivors of AMI, it was found that the prescription rate of ARBs increased from 2002 to 2015. Compared to patients on ACEIs, those on ARBs exhibited lower rates of all-cause mortality, cardiovascular death, and HHF. However, no significant differences were observed in adverse renal outcomes between the ACEI and ARB groups.

From 2002 to 2015, the prescription rate of ARBs increased. We observed that ARBs were preferred for RAS blockade in patients with AMI, particularly in patients with high-risk features, such as heart failure. Furthermore, our study demonstrated high tolerance to ARBs and prominent drug switches of initial ACEI users after a 3-month censor period (S2 Table).

Despite the debate of the ARB-MI paradox [14], our study found that ARB users showed lower all-cause mortality than ACEI users, which is different from the results of many clinical trials. It is broadly believed that ACEIs are more effective than ARBs in reducing CV outcomes, particularly in patients with AMI [14]. Most of the placebo control trials of ACEIs have shown significantly lower mortality and more favorable CV outcomes [15], which was not observed in the placebo control trials of ARBs [16]. One possible explanation is the "generation gap" between ACEI and ARB trials. ACEI trials were conducted in an era that did not widely use secondary prevention therapies and timely PCI, which may be the cause of the higher placebo event rate than in ARBs trials. Additionally, most head-to-head comparison trials have shown no significant difference between ACEI and ARB users [10, 11]. For example, in the OPTIMAAL trial, losartan use was associated with no significant difference in all-cause mortality but significantly higher CV death compared with captopril use [17], although some experts explained that the result could be related to the low dose of losartan used [10]. Additionally, the VALIANT trial showed no significant outcome difference between ACEI and ARB users [11]. Furthermore, results from two meta-analyses have yielded inconsistent findings, but these analyses did not focus on patients with acute myocardial infarction. [18, 19]; however, they both suggested ARBs to be as efficacious and safe as ACEIs, with the added advantage of higher tolerability when the study was restricted the trials after 2000. Most of these studies were conducted in Western countries and enrolled few Asians. Furthermore, studies conducted in Asia—most of which used data from the Korea Acute Myocardial Infarction Registry (KAMIR)—have shown inconsistent results [7, 20–23]. Because clinical trials, meta-analysis, and studies using real-world data have concluded that ARBs are noninferior to ACEIs, the treatment strategy could be different in Asian countries because Asian patients usually develop adverse side effects such as cough with ACEI therapy [24].

In our study, the incidence of AKI and renal replacement therapy in ACEI and ARB users was similar at 12- and 18-month follow-up periods. Although, to the best of our knowledge, no study has compared AKI risk in patients with AMI on ACEI and ARB therapies, one study based on registry data found that ACEI/ARB therapy may resulted in an increased risk of AKI, particularly in patients with an eGFR of >30 mL/min/1.73 m$^2$ compared with patients without ACEI/ARB therapy [25]. The incidence of AKI and end-stage renal failure requiring dialysis in a study was 1.22–1.47 and 0.39–0.51 per 1000 person-years, respectively, which were comparable with the normal population in Taiwan (0.43 per 1000 person-years in 2012) [26]. Although, a concern exists that renin angiotensin system inhibitor may cause adverse renal outcomes [27], particularly in CKD patients, our study showed no significant difference between ACEI and ARB users in terms of AKI and dialysis.

Our study differs from previous studies in several aspects. Firstly, only a few studies have focused on Asians. Most of these Asian trials used data from KAMIR, which investigated the clinical outcomes of ACEI and ARB users. However, the KAMIR data only evaluated patients' medications at discharge [7, 20, 23]. On the contrary, our study evaluated the actual medication usage within 3 months after discharge of myocardial infarction patients, which can provide a clearer reflection of the clinical outcomes difference for patients who actually used ACEIs and ARBs. Second, the results of the current studies were compared with those involving selected patient registries or clinical trial volunteers, although our data were derived from the daily clinical practice, which were closer to real-world data. The analysis based on generation was performed (2002–2005, 2006–2010, and 2011–2015), including bear metal stents and first- and second-generation drug eluting stents to increase the internal validity of the current study. Moreover, this study adopted several statistical tools to increase the robustness of the findings, including using PSM to reduce the selection bias, performing subgroup analysis to validate results across different groups, and conducting sensitivity analysis to ensure the medication switch did not affect the findings.

The current study is subject to a few limitations. First, we applied PSM to balance the major observable baseline covariates to reduce the selection bias; however, the residual confounding bias might still exist due to the observational study design. Second, the NHIRD does not provide information on blood pressure, factors associated with drug selection and dosage of a specific antihypertensive medication. Additionally, patient characteristics such as obesity and smoking status, allergy history, as well as laboratory data such as potassium levels, blood glucose, successful or unsuccessful PCI and actual ejection fraction, are also not available in the NHIRD. Third, we used the prescription claims to estimate medication adherence which was correlated with pill counts in other studies [28]. Still, actual drug adherence of individual was unable to obtain. Finally, the interruption or switching of drugs may have resulted in an overestimation of the effects of the respective drugs in the IIT analysis. However, similar findings were found when treating patients who switched their initial medication as censoring, as detailed in S3 and S4 Tables. Two analysis methods (IIT and as-treated analysis) were used to verify the data from NHIRD which were also employed in another database study [29].

The results of this study suggest that ARBs may be beneficial in reducing all-cause mortality and lowering the hospitalization rate for heart failure compared to ACEIs. Given the retrospective nature with several limitations, this study is hypothesis generating and further studies are needed to evaluate the role of ACEI versus ARB in Asian AMI patients.

## Supporting information

**S1 Table. Disease diagnostic coding and ATC code for medication.**
(PDF)

**S2 Table. Changes in the use of renin-angiotensin system inhibitors after 3 months of index date.**
(PDF)

**S3 Table. Incidence (per 1000 PM) and adjusted hazard risk of major outcomes for treatment efficiency based on as-treated analysis.**
(PDF)

**S4 Table. Incidence (per 1000 PM) and adjusted hazard risk of renal outcomes for treatment safety based on as-treated analysis.**
(PDF)

## Acknowledgments

The authors wish to thank Wallace Academic Editing for editing this manuscript.

## Author Contributions

**Conceptualization:** Chih-Wei Chen, Li-Nien Chien, Chun-Yao Huang.

**Data curation:** Wan-Ting Chen, Li-Nien Chien.

**Supervision:** Yi-Cheng Lin, Chun-Yao Huang.

**Writing – original draft:** Chih-Wei Chen, Chun-Wei Chang.

**Writing – review & editing:** Chih-Wei Chen.

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
