## [Decision Letter · Decision Letter 0]

25 Apr 2023

PONE-D-23-06580Comparison of Clinical Outcomes of Angiotensin Receptor Blockers with Angiotensin-Converting Enzyme Inhibitors in Patients with Acute Myocardial InfarctionPLOS ONE

Dear Dr. Huang,

Thank you for submitting your manuscript to PLOS ONE. After careful consideration, we feel that it has merit but does not fully meet PLOS ONE’s publication criteria as it currently stands. Therefore, we invite you to submit a revised version of the manuscript that addresses the points raised during the review process.

We look forward to receiving your revised manuscript.

Kind regards,

Timir Paul

Academic Editor

PLOS ONE

- 10.1097/HJH.0000000000000804

In your revision ensure you cite all your sources (including your own works), and quote or rephrase any duplicated text outside the methods section. Further consideration is dependent on these concerns being addressed.

Reviewers' comments:

Reviewer's Responses to Questions

**Comments to the Author**

1. Is the manuscript technically sound, and do the data support the conclusions?

Reviewer #1: Partly

Reviewer #2: Yes

2. Has the statistical analysis been performed appropriately and rigorously? 

Reviewer #1: Yes

Reviewer #2: Yes

3. Have the authors made all data underlying the findings in their manuscript fully available?

Reviewer #1: Yes

Reviewer #2: Yes

4. Is the manuscript presented in an intelligible fashion and written in standard English?

Reviewer #1: Yes

Reviewer #2: Yes

5. Review Comments to the Author

Reviewer #1: Dear author,

I appreciate your effort to conduct this study. The hypothesis you are exercising is helpful for day-to-day practice. I want to comment on a few things, which are stated below.

-Correction required in introduction. I left comments in a PDF file.

-It would be great to see some cost-effective analysis as well.

-54% of ACEI users switched to ARBs in a median time of 14 months. And relatively high mortality in the ACEI group compared to ARBs. Is it an ARB effect or other confounders like EF, the unsuccessful PCI, ICD, and door-to-device time?

-It's better to know the type of stents they use. A random sample from NHID Taiwan (2007- 2010) sample showed disproportionally high usage of BMS (65%)over DES(35%). https://journals.plos.org/plosone/article?id=10.1371/journal.pone.0179127

-Post MI Ejection fraction?

-How many have had improved EF?

-Use imaging during PCI if stent thrombosis is a common cause of mortality?

-How many vessels intervened?

-How many received ICD/CRT?

- Smoking history?

-All-cause mortality is high in the ACEI group. I would like to know the common cause of death in both groups.

-Door to balloon or device time?

-I wonder how reliable the final results are if we don't have ejection fraction, type of stents, how many and what vessel were revascularized, and success of PCI, primarily if your study population consist of ~65-70% STEMI.

Reviewer #2: I appreciate the efforts of the authors to answer this important clinical question as contradictory evidence has been shown by previous trials and studies. However, there are some recommendations and questions as below:

1: The discussion explains how is this study different from the previous studies. However, an article has recently been published in PLOS ONE by Jae-Geun Lee titled 'Impact of angiotensin-converting enzyme inhibitors versus angiotensin receptor blockers on clinical outcomes in hypertensive patients with acute myocardial infarction' which also answered this question using national registry in Asian population. Please address how is your study different from this study and what additional information does your study provide besides renal outcomes? This study by Lee JG gives contradictory results and states that ARBs are inferior to ACEI in AMI in Asian population.

2: Patients receiving both ACEI and ARBs were excluded from the study. However, it will be interesting to do a subgroup comparison of patients who received both ACEI and ARBs as compared to ACEI or ARB alone.

3: In Table 1, under 'Medication Use, yes' sub-heading, ACEI and ARB use is mentioned in both groups. Does this imply that these patients were receiving ACEI and ARB prior to AMI? If yes, it might be interesting to look into outcomes of patients who were continued on the same medication vs those who started it for the first time after AMI hospitalization.

4: Post-hoc analysis of CONSENSUS-II trial has shown that addition of aspirin may reduce ACEI benefit in AMI patients. Even though most of the AMI patients will be receiving aspirin, It will be clinically relevant to look into outcomes of patients on ACEI or ARB with vs without aspirin. Same can be done for beta blockers, CCB, or NOACS if data allows.

5: Some patients might have been prescribed ARBs if ACEI are contraindicated. If it's possible to see if patient was allergic to ACEI or if it was contraindicated, it will be relevant to tabulate outcomes in patients who received ARBs because they were allergic to ACEI vs those who were not allergic.

6: A comment should be added at the end whether you conclude that ARBs should be preferred in Asian population or further studies are required to make a recommendation. Studies have shown contradictory results even within Asian population eg. Jae-Geun Lee et al. It will be helpful if a study design is outlined with required variables which will be able to get more definite results.

7: ACEI/ARBS have shown to improve LVEF following AMI. It would've been intriguing to study the LVEF improvement but as mentioned in the study exact LVEF data isn't available in the database used. Since, hospitalization for heart failure is one of the main outcomes of the study. It will add to the strength of the study to compare the hospitalizations for heart failure with preserved vs reduced ejection fraction using respective ICD-10 codes.

6. PLOS authors have the option to publish the peer review history of their article (what does this mean?). If published, this will include your full peer review and any attached files.

Reviewer #1: **Yes: **Maheswara Satya Golla

Reviewer #2: No

---

## [Author Response · Author response to Decision Letter 0]

6 Jun 2023

Dear Editor, 

Thank you for considering our manuscript for publication in PLOS ONE. We appreciate the time and effort the academic editor and reviewers invested in evaluating our work. We have carefully reviewed the suggestions, and our responses are as follows. We also revised the manuscript by using track changes where the portion needs to be changed.

For Journal requirements:

Answers: 

We revised our manuscript to meet PLOS ONE's style requirements, including the title page, file naming, and author byline.

2. We noticed you have some minor occurrence of overlapping text with the following previous publication(s), which needs to be addressed:- 10.1097/HJH.0000000000000804

In your revision ensure you cite all your sources (including your own works), and quote or rephrase any duplicated text outside the methods section. Further consideration is dependent on these concerns being addressed.

Answers: 

Like the referenced article, our study utilized NHIRD to analyze the prognosis of ACEIs/ARBs in different groups (prior stroke and post-myocardial infarction). We also applied similar methods, such as propensity score matching, to reduce potential confounding factors inherent in retrospective data and employed two analysis methods (IIT and as-treated analysis) to address the issue regarding patients' actual medication adherence in the database. Therefore, we referenced the aforementioned article as reference no. 29 and addressed the limitation, as suggested by the reviewers.

Change: 

And we revise the description in the limitation section as follows: 

(page 19, line 19, discussion paragraph, limitation section) The current study is subject to a few limitations. First, we applied PSM to balance the major observable baseline covariates to reduce the selection bias; however, the residual confounding bias might still exist due to the observational study design. Second, the NHIRD does not provide information on blood pressure, factors associated with drug selection, and dosage of a specific antihypertensive medication. Additionally, patient characteristics such as obesity and smoking status, allergy history, and laboratory data such as potassium levels, blood glucose, successful or unsuccessful PCI, and actual ejection fraction are also not available in the NHIRD. Third, we used the prescription claims to estimate medication adherence which was correlated with pill counts in other studies. Still, actual drug adherence of individuals was unable to obtain. Finally, the interruption or switching of drugs may have resulted in an overestimation of the effects of the respective drugs in the IIT analysis. However, similar findings were found when treating patients who switched their initial medication as censoring, as detailed in S3 and S4 table. Two analysis methods (IIT and as-treated analysis) were used to verify the data from NHIRD, which were also employed in another database study (ref. 29)

3.Please include captions for your Supporting Information files at the end of your manuscript, and update any in-text citations to match accordingly. Please see our Supporting Information guidelines for more information: http://journals.plos.org/plosone/s/supporting-information

Answers: 

We included table captions for supplemental material at the end of our manuscript. Thank you for your reminder.

Reviewer 1

1. 54% of ACEI users switched to ARBs in a median time of 14 months. And relatively high mortality in the ACEI group compared to ARBs. Is it an ARB effect or other confounders like EF, the unsuccessful PCI, ICD, and door-to-device time?

Answers: 

Thank you for the comment. The data of actual left ventricular ejection fraction, successful or unsuccessful PCI, and door-to-device time were unavailable in our NHIRD. Alternatively, we used International Classification of Diseases (ICD) codes to define underlying comorbidities of heart failure and employed propensity score matching to reduce the effect of confounding factors. We addressed the limitation in the discussion.

Change: 

(page 20, line 4, discussion paragraph, limitation section) Additionally, patient characteristics such as obesity and smoking status, allergy history, and laboratory data such as potassium levels, blood glucose, successful or unsuccessful PCI, and actual ejection fraction are also not available in the NHIRD.

2. It's better to know the type of stents they use. A random sample from NHID Taiwan (2007- 2010) sample showed disproportionally high usage of BMS(65%) over DES(35%). 

https://journals.plos.org/plosone/article?id=10.1371/journal.pone.0179127

Answers: 

Thank you for the comment. Indeed, due to the reimbursement conditions of the National Health Insurance in Taiwan, the use of bare metal stents (BMS) has been relatively higher in the past (Acta Cardiol Sin 2014;30:553-564), but the utilization rate of drug-eluting stents (DES) has gradually increased in recent years. Our study spanned several time periods (2002-2005, 2006-2010, 2011-2015), and users of angiotensin receptor blockers (ARBs) consistently associated with better all-cause mortality outcomes compared to users of angiotensin-converting enzyme inhibitors (ACEIs) across different time periods. We recognized that we did not consider the information regarding the type of stent; however, the results should not be influenced.

3. Post MI Ejection fraction?

-How many have had improved EF?

-Use imaging during PCI if stent thrombosis is a common cause of mortality?

-How many vessels intervened?

-How many received ICD/CRT?

-Smoking history?

-Door to balloon or device time?

Answers: 

Thank you for the comment, as mentioned in the previous point, the data mentioned above were not available in our NHIRD. Therefore, we have modified the description in the limitation section.

Change: 

(page 20, line 4, discussion paragraph, limitation section) Additionally, patient characteristics such as obesity and smoking status, allergy history, and laboratory data such as potassium levels, blood glucose, successful or unsuccessful PCI, and actual ejection fraction are also not available in the NHIRD.

4.All-cause mortality is high in the ACEI group. I would like to know the common cause of death in both groups.

Answers: 

Thank you for your comment. Our study aims to compare the all-cause mortality and cardiovascular outcomes between users of ACEIs and ARBs in patients with acute myocardial infarction. Thus, we did not report the data on non-cardiovascular death.

5. I wonder how reliable the final results are if we don't have ejection fraction, type of stents, how many and what vessel were revascularized, and success of PCI, primarily if your study population consist of ~65-70% STEMI.

Answers: 

Thank you for the suggestions. Indeed, in this retrospective cohort study, we still have several limitations and potential biases, which we have addressed in the limitation section. We, therefore, have employed various methods to mitigate confounding factors in our study, such as using propensity score matching (PSM) and employing different analytical approaches (intention-to-treat and as-treated). Additionally, we utilized the medication possession ratio (MPR) to confirm that our analysis focused on actual medication users of ACEIs or ARBs among AMI patients. The results were also consistent in the sensitivity analyses. We were more confident that this study still holds a certain reference value.

Reviewer 2

1.The discussion explains how is this study different from the previous studies. However, an article has recently been published in PLOS ONE by Jae-Geun Lee titled 'Impact of angiotensin-converting enzyme inhibitors versus angiotensin receptor blockers on clinical outcomes in hypertensive patients with acute myocardial infarction' which also answered this question using national registry in Asian population. Please address how is your study different from this study and what additional information does your study provide besides renal outcomes? This study by Lee JG gives contradictory results and states that ARBs are inferior to ACEI in AMI in Asian population.

Answers: 

Thank you for your comment. The reviewer mentioned that the article by Lee JG, similar to other studies using the KAMIR database, analyzes long-term outcomes based on medication usage at discharge for AMI patients. However, medication usage at discharge does not necessarily indicate continued medication usage after discharge. Therefore, we only included the patients whose medication possession ratio (MPR) was > 40% to ensure they had a certain level of ACEIs or ARBs usage following AMI discharge. In the revised manuscript, we have added the article suggested by the reviewer as the 23rd reference in our manuscript and revise the description in the discussion section as follows: 

Change: (page 18, line 7, discussion paragraph) Most of these studies were conducted in Western countries and enrolled a few Asians. Furthermore, studies conducted in Asia - most of which used data from the Korea Acute Myocardial Infarction Registry (KAMIR) - have shown inconsistent results [7, 20-23].

(page 19, line 5, discussion paragraph) Our study differs from previous studies in several aspects. Firstly, only a few studies have focused on Asians. Most of these Asian trials used data from KAMIR, which investigated the clinical outcomes of ACEI and ARB users. However, the KAMIR data only evaluated patients' medications at discharge [7, 20, 23]. On the contrary, our study evaluated the actual medication usage within 3 months after discharge of myocardial infarction patients, which can provide a clearer reflection of the clinical outcomes difference for patients who actually used ACEIs and ARBs

2. Patients receiving both ACEI and ARBs were excluded from the study. However, it will be interesting to do a subgroup comparison of patients who received both ACEI and ARBs as compared to ACEI or ARB alone.

Answers:

Thank you for your comment. In our study, the number of patients who concurrently received ACEIs and ARBs was only 1370 individuals (2.6% of the total population). With such a small number, conducting further subgroup analysis may be challenging. On the other hand, findings from the ONTARGET trial (N Engl J Med 2008; 358:1547-1559) suggest that the combination of ACEIs and ARBs may be linked to a higher occurrence of adverse events in patients with cardiovascular risk, without any discernible increase in benefits. 

3. In Table 1, under 'Medication Use, yes' sub-heading, ACEI and ARB use is mentioned in both groups. Does this imply that these patients were receiving ACEI and ARB prior to AMI? Yes. If yes, it might be interesting to look into outcomes of patients who were continued on the same medication vs those who started it for the first time after AMI hospitalization.

Answers: 

We appreciate your opinion. Yes, in Table 1, the medication use represents the drugs taken by the patients before the index AMI. In fact, in our subgroup analysis depicted in Figure 3, we specifically examined the impact of ACEIs or ARBs usage before the index AMI on mortality outcomes after the index AMI. Based on our analysis results, it appears that there is no significant effect on ARBs users associated with lower mortality than ACEIs users after AMI.

4. Post-hoc analysis of CONSENSUS-II trial has shown that addition of aspirin may reduce ACEI benefit in AMI patients. Even though most of the AMI patients will be receiving aspirin, It will be clinically relevant to look into outcomes of patients on ACEI or ARB with vs without aspirin. Same can be done for beta blockers, CCB, or NOACS if data allows.

Answers: 

We appreciate your opinion. In our study population (Table 1), nearly 90% of the patients were using aspirin after AMI (91.8% in ACEIs users, 88.9% in ARBs users). Due to the small proportion of patients not using aspirin, conducting further subgroup analysis may not be appropriate from a statistical perspective. In Figure 3, we performed a subgroup analysis based on using beta-blockers or statins to assess their impact on our outcomes, but the results remained unchanged. The use of NOACs accounted for less than 1% of the analyzed population, and we did not conduct further analysis for CCBs as they are not essential medications after AMI.

5.Some patients might have been prescribed ARBs if ACEI are contraindicated. If it's possible to see if patient was allergic to ACEI or if it was contraindicated, it will be relevant to tabulate outcomes in patients who received ARBs because they were allergic to ACEI vs those who were not allergic.

Answers: 

Thank you for your comment. Due to the nature of our study being a retrospective cohort study using data from NHIRD, we did not have access to information regarding allergy history and the factors considered in the selection of ACEIs or ARBs. We would like to add it to the limitation regarding factors associated with medication switching that were unable to obtain in this study. 

Change: 

(page 20, line 2, discussion paragraph, limitation section) Second, the NHIRD does not provide information on blood pressure, factors associated with drug selection and dosage of a specific antihypertensive medication. Additionally, patient characteristics such as obesity and smoking status, allergy history, and laboratory data such as potassium levels, blood glucose, successful or unsuccessful PCI, and actual ejection fraction are also not available in the NHIRD.

6. A comment should be added at the end whether you conclude that ARBs should be preferred in Asian population or further studies are required to make a recommendation. Studies have shown contradictory results even within Asian population eg. Jae-Geun Lee et al. It will be helpful if a study design is outlined with required variables which will be able to get more definite results.

Answers: 

We appreciate your opinion. We have made the following modifications to the "conclusion" section in the discussion

Change: 

(page 20, line 16, discussion paragraph, conclusion section) Our study may have clinical implications for Asians after MI. Despite the contradictory results observed in some studies focusing on the Asian population, our analysis only included the patients whose MPR ≥ 40% to ensure the patients were actual medication users of ACEIs or ARBs, suggests that ARBs may be beneficial in reducing all-cause mortality and lowering the hospitalization rate for heart failure compared to ACEIs.

7.ACEI/ARBS have shown to improve LVEF following AMI. It would've been intriguing to study the LVEF improvement but as mentioned in the study exact LVEF data isn't available in the database used. Since, hospitalization for heart failure is one of the main outcomes of the study. It will add to the strength of the study to compare the hospitalizations for heart failure with preserved vs reduced ejection fraction using respective ICD-10 codes.

Answers:

We appreciate your opinion. However, as mentioned earlier, the lack of LVEF data in the NHIRD is a limitation of our study. Using ICD-10 to differentiate hospitalization events for heart failure into preserved or reduced ejection fraction is indeed a good approach. However, since the conversion of NHIRD reporting data from ICD-9 to ICD-10 began in 2016, and our study data spans from 2002 to 2015, hospitalizations for heart failure cannot be obtained with ICD-10 data, making it difficult to distinguish preserved or reduced ejection fraction. Nonetheless, we still appreciate the reviewer for providing us with such valuable research method suggestions.

We hope that the revised manuscript will meet the expectations of both the reviewers and yourself and await the decision concerning publication of PLOS ONE. 

Sincerely,

On behalf of all authors,

Li-Nien Chien

Institute of Health and Welfare Policy, College of Medicine, National Yang Ming Chiao Tung University, Taipei, Taiwan

Email address: linien.chien@nycu.edu.tw (LNC)

Chun-Yao Huang

Division of Cardiology, Department of Internal Medicine and Cardiovascular Research Center, Taipei Medical University Hospital, Taipei, Taiwan

Email address: cyhuang@h.tmu.edu.tw (CYH)

---

## [Decision Letter · Decision Letter 1]

1 Aug 2023

PONE-D-23-06580R1Comparison of Clinical Outcomes of Angiotensin Receptor Blockers with Angiotensin-Converting Enzyme Inhibitors in Patients with Acute Myocardial InfarctionPLOS ONE

Dear Dr. Huang,

Thank you for submitting your manuscript to PLOS ONE. After careful consideration, we feel that it has merit but does not fully meet PLOS ONE’s publication criteria as it currently stands. Therefore, we invite you to submit a revised version of the manuscript that addresses the points raised during the review process.

The authors responded al the queries appropriately and incorporated the changes in the manuscript with added information to the limitations as per reviewers' suggestions. Although there are several limitations given the retrospective in nature of the data I believe this study has merit to publish. There are several strengths of the study such as  larger sample size, propensity match analyses and usage of medication possession ratio (MPR) was > 40% compared to other studies. The propensity match would reduce the some of the confounders although not entirely. Due to retrospective database there are several inherent bias, confounders and missing data/variables as outlined in the limitations section of the revised manuscript. I agree with the reviewer that there are certain data that are important for this type of study questions such as LVEF, multivessel  disease, shock etc. Unfortunately these information is missing. There are subgroup analyses that is helpful such as stent type with different time periods and comparison. Given the conflicting results with few studies in Asian population, publishing this study is worthy. However, I suggest authors to add the following statement and minor revision as follows:

My suggestions:

Conclusion:

I will delete this sentence and put  in the discussion section as appropriate  if not stated “Our study may have clinical implications for Asians after MI. Despite the contradictory results 340 observed in some studies focusing on the Asian population [7, 20, 23], our analysis only included the 341 patients whose MPR ≥40% to ensure the patients were actual medication users of ACEIs or ARBs”

I will make the conclusion statement simple:

The results of this study suggest that ARBs may be beneficial in reducing all-cause mortality and **lowering the hospitalization rate for heart failure compared to ACEIs.**

**In the conclusion I will add the following statement**

Given the retrospective nature with several limitations, this study is hypothesis generating and further studies are needed to evaluate the role of ACEI versus ARB in Asian AMI patients

Please submit your revised manuscript by Sep 15 2023 11:59PM. If you will need more time than this to complete your revisions, please reply to this message or contact the journal office at plosone@plos.org. Please include the following items when submitting your revised manuscript:A rebuttal letter that responds to each point raised by the academic editor and reviewer(s). You should upload this letter as a separate file labeled 'Response to Reviewers'.A marked-up copy of your manuscript that highlights changes made to the original version. You should upload this as a separate file labeled 'Revised Manuscript with Track Changes'.An unmarked version of your revised paper without tracked changes. You should upload this as a separate file labeled 'Manuscript'.If applicable, we recommend that you deposit your laboratory protocols in protocols.io to enhance the reproducibility of your results. Protocols.io assigns your protocol its own identifier (DOI) so that it can be cited independently in the future. For instructions see: https://journals.plos.org/plosone/s/submission-guidelines#loc-laboratory-protocols. Additionally, PLOS ONE offers an option for publishing peer-reviewed Lab Protocol articles, which describe protocols hosted on protocols.io. Read more information on sharing protocols at https://plos.org/protocols?utm_medium=editorial-email&utm_source=authorletters&utm_campaign=protocols.

We look forward to receiving your revised manuscript.

Kind regards,

Timir Paul

Academic Editor

PLOS ONE

Journal Requirements:

Additional Editor Comments:

See recommendations above with similar suggestions as below 

I suggest authors to add the following statement and minor revision as follows:

My suggestions:

Conclusion:

I will delete this sentence and put  in the discussion section as appropriate  if not stated “Our study may have clinical implications for Asians after MI. Despite the contradictory results 340 observed in some studies focusing on the Asian population [7, 20, 23], our analysis only included the 341 patients whose MPR ≥40% to ensure the patients were actual medication users of ACEIs or ARBs”

I will make the conclusion statement simple:

The results of this study suggest that ARBs may be beneficial in reducing all-cause mortality and **lowering the hospitalization rate for heart failure compared to ACEIs.**

**In the conclusion I will add the following statement**

Given the retrospective nature with several limitations, this study is hypothesis generating and further studies are needed to evaluate the role of ACEI versus ARB in Asian AMI patients

Reviewers' comments:

Reviewer's Responses to Questions

**Comments to the Author**

1. If the authors have adequately addressed your comments raised in a previous round of review and you feel that this manuscript is now acceptable for publication, you may indicate that here to bypass the “Comments to the Author” section, enter your conflict of interest statement in the “Confidential to Editor” section, and submit your "Accept" recommendation.

Reviewer #1: All comments have been addressed

Reviewer #2: All comments have been addressed

2. Is the manuscript technically sound, and do the data support the conclusions?

Reviewer #1: No

Reviewer #2: Yes

3. Has the statistical analysis been performed appropriately and rigorously? 

Reviewer #1: Yes

Reviewer #2: I Don't Know

4. Have the authors made all data underlying the findings in their manuscript fully available?

Reviewer #1: Yes

Reviewer #2: Yes

5. Is the manuscript presented in an intelligible fashion and written in standard English?

Reviewer #1: Yes

Reviewer #2: Yes

6. Review Comments to the Author

Reviewer #1: ACEI inhibitors are very cost-effective medication in the post-MI heart failure population. As I mentioned in my initial review, I would like more data to convince myself that ARB is better than ACEI, especially mortality in heart failure. The propensity score is helpful if you include all the variables. How does the propensity score rule out the effect of essential variables if you don't include them in the analysis?

Reviewer #2: (No Response)

7. PLOS authors have the option to publish the peer review history of their article (what does this mean?). If published, this will include your full peer review and any attached files.

Reviewer #1: **Yes: **Maheswara Satya GR Golla

Reviewer #2: No

---

## [Author Response · Author response to Decision Letter 1]

2 Aug 2023

Dear Editor, 

Thank you once again for your response and consideration to publish our article in the PLOS ONE journal. We truly value the time and effort invested by the academic editor and reviewers in evaluating our work. We have thoroughly reviewed the suggestions and our responses are as follows.

Reviewer 1

I suggest authors to add the following statement and minor revision as follows:

My suggestions:

Conclusion:

I will delete this sentence and put in the discussion section as appropriate if not stated “Our study may have clinical implications for Asians after MI. Despite the contradictory results observed in some studies focusing on the Asian population [7, 20, 23], our analysis only included the patients whose MPR ≥40% to ensure the patients were actual medication users of ACEIs or ARBs”

I will make the conclusion statement simple:

The results of this study suggest that ARBs may be beneficial in reducing all-cause mortality and lowering the hospitalization rate for heart failure compared to ACEIs.

In the conclusion I will add the following statement

Given the retrospective nature with several limitations, this study is hypothesis generating and further studies are needed to evaluate the role of ACEI versus ARB in Asian AMI patients

Answers: 

We greatly appreciate your valuable input. Based on the reviewer's suggestion, we have revised our conclusion as follows.

Change: 

(page 20, line 14, discussion paragraph, conclusion section) The results of this study suggest that ARBs may be beneficial in reducing all-cause mortality and lowering the hospitalization rate for heart failure compared to ACEIs. Given the retrospective nature with several limitations, this study is hypothesis generating and further studies are needed to evaluate the role of ACEI versus ARB in Asian AMI patients.

We hope that the revised manuscript will meet the expectations of both the reviewers and yourself and await the decision concerning publication of PLOS ONE. 

Sincerely,

On behalf of all authors,

Li-Nien Chien

Institute of Health and Welfare Policy, College of Medicine, National Yang Ming Chiao Tung University, Taipei, Taiwan

Email address: linien.chien@nycu.edu.tw (LNC)

Chun-Yao Huang

Division of Cardiology, Department of Internal Medicine and Cardiovascular Research Center, Taipei Medical University Hospital, Taipei, Taiwan

Email address: cyhuang@h.tmu.edu.tw (CYH)

---

## [Editor Report · Decision Letter 2]

4 Aug 2023

Comparison of Clinical Outcomes of Angiotensin Receptor Blockers with Angiotensin-Converting Enzyme Inhibitors in Patients with Acute Myocardial Infarction

PONE-D-23-06580R2

Dear Dr. Huang,

We’re pleased to inform you that your manuscript has been judged scientifically suitable for publication and will be formally accepted for publication once it meets all outstanding technical requirements.

Kind regards,

Timir Paul

Academic Editor

PLOS ONE

Additional Editor Comments (optional):

All comments have been addressed.
---

## [Editor Report · Acceptance letter]

5 Sep 2023

PONE-D-23-06580R2 

Comparison of Clinical Outcomes of Angiotensin Receptor Blockers with Angiotensin-Converting Enzyme Inhibitors in Patients with Acute Myocardial Infarction 

Dear Dr. Huang:

I'm pleased to inform you that your manuscript has been deemed suitable for publication in PLOS ONE. Congratulations! Your manuscript is now with our production department. 

Kind regards, 

on behalf of

Dr. Timir Paul 

Academic Editor

PLOS ONE